# Turkey Oak (*Quercus cerris* L.) Resilience to Climate Change: Insights from Coppice Forests in Southern and Central Europe

Michaela Šimková [1], Stanislav Vacek [1], Václav Šimůnek [1,*], Zdeněk Vacek [1], Jan Cukor [1,2], Vojtěch Hájek [1], Lukáš Bílek [1], Anna Prokůpková [1], Igor Štefančík [3], Zuzana Sitková [3] and Ivan Lukáčik [4]

1 Department of Silviculture, Faculty of Forestry and Wood Sciences, Czech University of Life Sciences Prague, Kamýcká 129, CZ-165 00 Prague-Suchdol, Czech Republic
2 Forestry and Game Management Research Institute, Strnady 136, CZ-252 02 Jíloviště, Czech Republic
3 National Forest Centre, Forest Research Institute, T. G. Masaryka 2175/22, 960 01 Zvolen, Slovakia
4 Department of Silviculture, Faculty of Forestry, Technical University in Zvolen, T.G. Masaryka 24, 960 53 Zvolen, Slovakia
* Correspondence: simunekv@fld.czu.cz

**Abstract:** Turkey oak (*Quercus cerris* L.) is a thermophilic oak species that is gaining importance in the context of ongoing climate change because of its better resistance to climatic extremes and drier conditions. Therefore, this article focuses on Turkey oak's role and growth properties in the coppice forests of Southern Europe (Italy, Bulgaria) compared to similar site conditions in Central Europe (Slovakia, Czechia). The aims are to evaluate the basic dendrometry indicators, stand biodiversity, growth dynamics, and the effect of climatic factors on tree-ring increment on specific site chronologies. We found that the tree density in coppices of 50–60 years varied between 475 and 775 trees ha$^{-1}$, and the stand volume ranged from 141 to 407 m$^3$ ha$^{-1}$. The complex stand diversity of all plots ranged from a monotonous to uniform structure. The size of tree-ring growth was closely related to indicators of stand density. The lowest influence of climatic factors on tree-ring growth was found in sites in Italy and the highest in Slovakia. The primary limiting factor for growth was the lack of precipitation during the growing season, especially in June and July. In contrast, temperature had a marginal effect on radial growth compared to precipitation. The radial growth in research plots in Southern Europe goes through longer 6 to 8-year growth cycles, and in Central European sites, it goes through shorter cycles of 2.4 to 4.8 years, which confirms better growth conditions in this region. The studied coppice stands exhibit a stable reaction to climate change. Yet, regarding cyclical growth, the Central European stands benefit from an advantageous climate and grow better than in Southern Europe. As part of the changing environmental conditions, Turkey oak is becoming an important tree species that can achieve high production potential even in drier habitats due to its regeneration characteristic as coppice and may play a critical role in its northerly introduction in Europe.

**Keywords:** dendrochronology; cyclical dynamics; stand structure; biodiversity; timber production

## 1. Introduction

Forest ecosystems are experiencing ongoing climate change worldwide, and the changes are evident, especially in European forests [1–4]. Standard forest management practices appear unsuitable for providing sustainable and stable timber production and ecosystem services such as carbon sequestration [5–7], improved water dynamics in forest stands, drought mitigation [8,9], biodiversity conservation [10], and many other ecosystem functions. Therefore, forest managers are looking for alternatives to standard management practices, which have existed for decades or even centuries, aiming to provide essential benefits, such as timber production, thereby generating income for forest owners in the first place [11–13].

Coppice forests are an alternative to standard forest management practices. However, it is an almost forgotten management method in some European regions [14]. Coppicing





represents the oldest form of systematic and sustainable use of these forests [15,16]. It is a very flexible system that requires low energy consumption and skilled labor and has adapted to the needs of rural societies that provide logs for fuel, charcoal, agriculture, and small businesses [17,18]. The owners or users of these forests build on local ecological knowledge to help maintain and increase the resilience of this social-ecological forest management system [19–21].

The management is based on the stump sprouting of some tree species after felling [22]. Coppice is established by clonal stems forming interconnected groups or predominantly multi-stemmed clusters of individuals that have arisen by vegetative propagation [23,24]. The rotation period of the stand is usually short (about 15–30 years) and, therefore, significant structural changes occur during coppicing [25]. Ancient coppices are a specific type of habitat that reflects long-term human influence and contains high species biodiversity [26]. They preserve local tree ecotypes and, in some locations, are the only remnants of original trees with a natural species composition, even though the stand structure has changed [27]. Coppices show enormous variability and adaptability in the tree and herb layers and in their growth processes [28–30]. This management method preserves the biodiversity of plants, including rare species of ground flora [31], and is, therefore, of considerable interest to nature conservation [32–34].

Growth of coppice begins when a single-trunk tree is felled. Then, multiple shoots start to sprout, forming a multi-stemmed tree. Repeated felling results in multi-stemmed trees. Several of these multi-stemmed trees in one area create a coppice forest [35]. Compared to seed regeneration, coppice's initial growth is much faster [36] thanks to the well-developed root system of harvested trees [37]. A sprout can grow up to 1 m per year, depending on many factors, such as tree species, habitat conditions, the stump's age, or the timing of the logging operation [38].

The coppice stands are characterized by rapid changes in thermal, light, and hydrological regimes [24]. The dynamics of regenerative forests thus offer a highly heterogeneous environment within a relatively small area [39]. As a result of regular harvesting interventions, all phases of forest succession occur periodically [35,40], enabling the coexistence of species with different strategies—light-demanding and shade-tolerant [41]. In the initial stages of the coppice cycle, open areas benefit light-demanding species. As the stand density increases and the canopy closes, it limits the ground vegetation growth [35]. These stands are dense, but as competition increases, gaps are formed in later stages due to the death of some sprouts or logging operations [42,43]. In addition to ground vegetation, coppice management influences lichens, fungi, beetles, saproxylic insects, [24,44], and birds [45].

Currently, coppice forests are most widespread throughout the Mediterranean, covering 23 million ha [35,46,47]. For example, in Italy, coppice forests cover 3.663 million hectares [35], and of these, both evergreen and deciduous forests of *Quercus* spp. encompass an area of approximately 1.6 million ha [17]. In Slovakia, coppices now cover 110,000 ha, compared to 1950 when they covered 196,000 ha, so there is evidence of a decreasing trend of this management practice [48]. The same trend is evident in Czechia, where most of the coppice forests were transformed into high forests [49] (a forest of generative origin with a usual production period of at least 100 years) [50]. In Czechia, coppice forests covered only 109,900 ha in 2013, whereas in 1845, they covered 1,457,400 ha [49]. These transformations from coppice forests are also evident in other European countries, especially since the second half of the 20th century when this forest management declined significantly [41].

Coppice forests in the past were mainly preserved in the form of stumps, which were either transferred to high forests or left untouched [35,39,51]. The primary goal of converting coppices to a high forest is to restore the original physiognomic, structural, and spatial diversity of close-to-nature stands [52,53]. Coppices can be converted to high forest in basically two ways: passively by aging (without intervention) or actively by thinning [54]. On the other hand, the current trend is the opposite and is more inclined to coppice forest conservation and restoration in many areas [24]. Among the countries that

still actively use coppice forest management are, for example, Romania, Bulgaria, Austria, France, Italy, and Spain [55]. In the last two decades, there has been renewed interest in coppice regarding their ecological functions and the provision of numerous ecosystem services [17,56]. In addition, regeneration in coppice is less damaged by game than artificial planting due to the increasing numbers of ungulates in Europe [57,58]. Interest in coppice forest management has also increased within the last decade due to the rising importance of their resilience to climate change [56].

Traditionally, coppice forest stands consist of broadleaved tree species with a high potential for coppice management [24]. Oaks are among the most suitable species utilized for this purpose, such as native sessile oak (*Quercus petraea*) [33]. Ongoing climate change creates suitable conditions for thermophilic oak species, including Turkey oak (*Quercus cerris* L.), which originally expanded mainly to Southeastern Europe but currently spread to Central Europe. Existing stands of Turkey oak, principally coppices in Europe, are most likely the result of 5000 years of human activity on these stands [59,60]. Coppices of Turkey oak stands in the hills of Italy cover 675,532 ha (18.4%) of the forest area [16]. Coppices in Italy have proven to be the most important cultivation system, widespread primarily in private deciduous forests [61]. In Bulgaria, Turkey oak stands cover an area of 258,400 ha (7.0%) [62], and Turkey oak is also common in Slovakia, where it covers 50,773 ha (2.6%) [63]. Moreover, thermophilic tree species, such as Turkey oak, can play a significant role in the context of ongoing climate change because they can better withstand climatically demanding and drier conditions [4,64]. The synergism of these two factors can be helpful for its successful introduction into new areas outside its natural range of distribution at present, when many native tree species, such as Norway spruce (*Picea abies* [L.] Karst.), are experiencing a large-scale decline in Europe [65].

Turkey oak growth and its reaction to ongoing climate change in Central and Southern Europe have not yet been described in detail. Closer studies of this type are necessary for understanding the adaptability of this tree species to climate change and the possibility of introduction outside its native areas. However, for the first step in determining growth processes, it is crucial to thoroughly analyze the stand structure (diversity, horizontal, and vertical structure) and production parameters (tree density, stocking, stand volume, etc.), which significantly affect the response of trees to climate change and cyclical events [4,66]. Therefore, the main objectives of this paper are (i) to evaluate the production potential, structure, and biodiversity of Turkey oak coppices in research sites in Italy, Bulgaria, Slovakia, and Czechia, (ii) determine growth conditions, cyclical dynamics, and the effect of climate factors (temperature and precipitation) on tree-ring growth in site chronologies in Southern and Central Europe, (iii) analyze the relationship between stand structure, production parameters, and tree-ring growth, and finally (iv) evaluate the growth dynamics during ongoing climate change.

## 2. Methodology

### 2.1. Study Area

Areas of interest in Southern and Central Europe (Italy, Bulgaria, Slovakia, and Czechia) include monospecific coppices of Turkey oak. In Italy and Bulgaria, these are foothills to mountainous locations, and in Slovakia and Czechia, they are lowlands. In individual countries, coppices aged 50–60 years were selected in areas where this silvicultural method has a thousand-year tradition. Coppicing is a silvicultural system based on systematically repeating the vegetative regeneration of sprouts, whereas high forest is regenerated generatively and the production period is at least 100 years [50]. Forest management on PRPs is based on individual thinning with respect to health status and target diameter at breast height (DBH). All studied stands belong to the association *Quercion pubescenti-petraeae*. The location of the permanent research plots (PRPs) is shown in Figure 1, and a basic overview of PRPs in Table 1. Long-term annual air temperature ranges from 8.8 to 13.7 °C, and the average sum of precipitation is 497–1022 mm in the study areas (Table 2). According to the worldwide Köppen classification, PRPs in Czechia and Slovakia

belong to the climate categorization Cfb—temperate oceanic climate with cool summers, mild winters, and a relatively narrow annual temperature range; PRPs in Bulgaria belong to Dfb—warm summer, humid continental climate, with substantial seasonal temperature differences; PRPs in Italy belong to Csa—hot summer, Mediterranean climate, with dry summers and mild, wet winters.

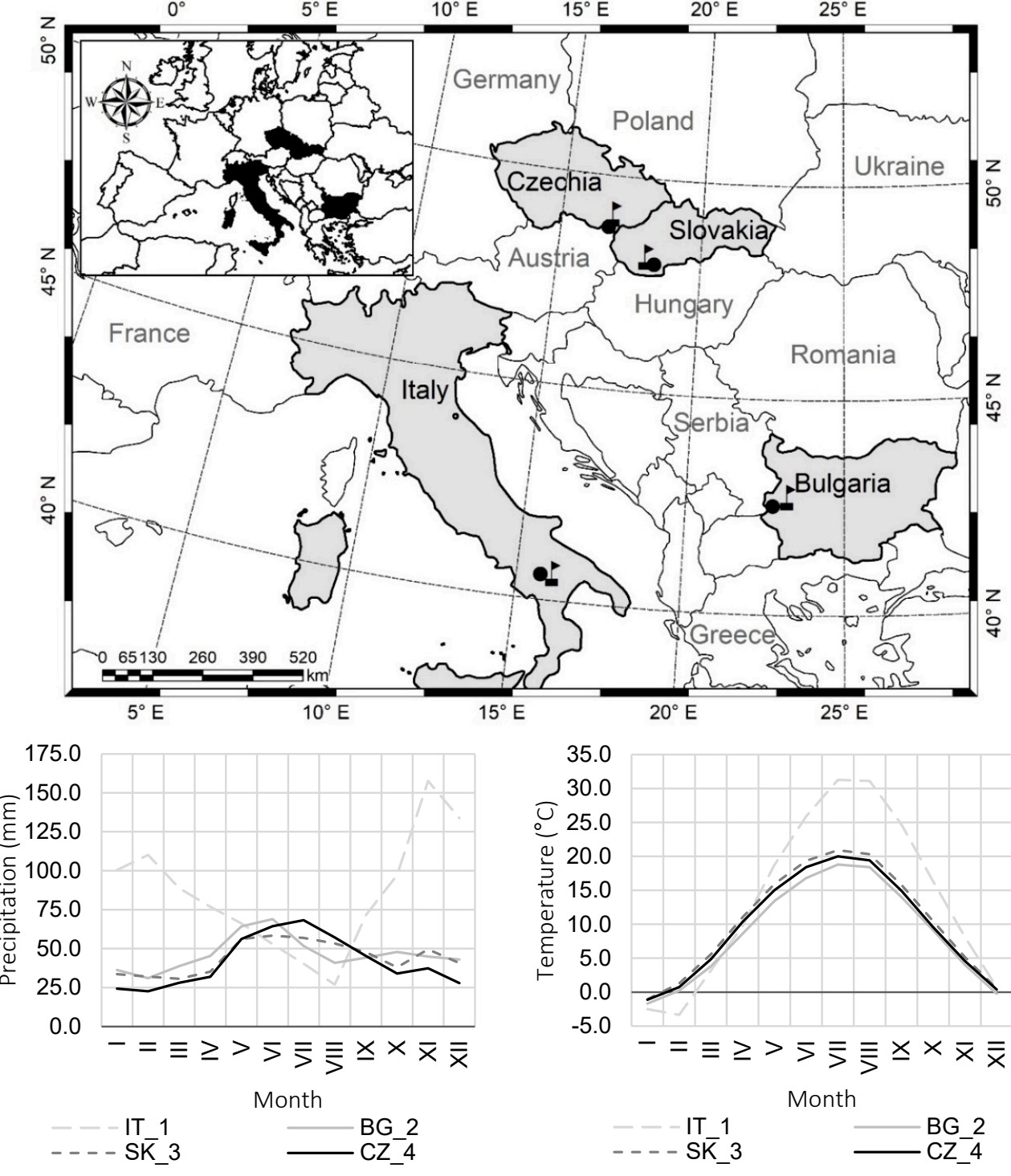

**Figure 1.** Localization of permanent research plots in coppices of Turkey oak in areas of interest in Europe (**up**); the precipitation (**down left**) and air temperature (**down right**) during the calendar year from I (January) to XII (December) in Italy (IT_1), Bulgaria (BG_2), Slovakia (SK_3), and Czechia (CZ_4).

**Table 1.** Overview of basic site and stand characteristics of Turkey oak coppices in areas of interest in Europe.

| Plot Name | Country | Coordinates | Altitude | Exposition | Slope | Geological Bedrock | Soil | Age | DBH | Height | Stand Volume |
|---|---|---|---|---|---|---|---|---|---|---|---|
| | | | (m) | | (°) | | | | (cm) | (m) | (m³ ha⁻¹) |
| IT_1 | Italy | 40°32′53.025″ N 15°43′38.775″ E | 1000 | W | 4.3 | limestone, marl | rendzina | 60 | 31 | 22 | 407 |
| BG_2 | Bulgaria | 42°31′13.619″ N 22°33′4.277″ E | 1060 | SW | 5.7 | sandstone | cambisol | 50 | 19 | 16 | 141 |
| SK_3 | Slovakia | 48°4′14.677″ N 18°21′58.449″ E | 220 | SE | 2.9 | loess clay | cambisol | 60 | 29 | 21 | 275 |
| CZ_4 | Czechia | 48°44′31.877″ N 16°47′35.951″ E | 200 | S | 0.0 | marl | cambisol | 60 | 28 | 22 | 342 |

Notes: W—west, SW—southwest, SE—southeast, S—south, DBH—diameter breast height.

**Table 2.** Overview of basic meteorological characteristics of research areas.

| Plot | Meteo. Station Name | GPS of Meteo. Station | Station Altitude (m a.s.l.) | Distance to Plot (km) | Annual Temperature (°C) | Seasonal Temperature (°C) | Annual Precipitation (mm) | Seasonal Precipitation (mm) |
|---|---|---|---|---|---|---|---|---|
| IT_1 | Abriola | 40°31′8″ N 15°47′38″ E | 1225 | 6.5 | 13.7 | 26.3 | 1022 | 258 |
| BG_2 | Divlya | 42°28′43″ N 22°41′34″ E | 720 | 12.5 | 8.8 | 16.3 | 552 | 268 |
| SK_3 | Hurbanovo | 47°52′00″ N 18°12′00″ E | 115 | 25.8 | 10.4 | 18.5 | 532 | 273 |
| CZ_4 | Lednice | 48°47′35″ N 16°47′58″ E | 177 | 5.7 | 9.8 | 17.6 | 497 | 291 |

## 2.2. Data Collection

From 2021 to 2022, four PRPs with a size of 25 × 25 m (0.0625 ha) were inventoried. The structure of the tree layer was measured using FieldMap technology (IFER-Monitoring and Mapping Solutions Ltd., Jílové u Prahy, Czech Republic). Each stem was regarded as an individual tree, for both single stems and polycormons (stem with more shoots). The diameter at breast height, position of all individuals with a DBH ≥ 4 cm, total height, height of the green crown base, and crown projection area (at least in four mutually perpendicular directions) were measured. The height of the live crown base was measured at the point where branches formed a continuous whorl of a crown. The crown radii in four directions were measured at a right angle to each other through the centroid of the crown by FieldMap hardware [67]. Boundary trees with more than half of their DBH lying inside a PRP were included. The diameter at breast height was measured with a Mantax Blue caliper (Haglöf, Långsele, Sweden) with an accuracy of 1 mm, while DBH was averaged from two measurements. Individual tree height and the height of the live crown base were recorded with a Vertex laser hypsometer (Haglöf, Långsele, Sweden) with an accuracy of 0.1 m.

For the analysis of the radial growth of Turkey oak, core samples were obtained from the trees with a Pressler auger (Haglöf, Långsele, Sweden) at a height of 1.3 m in the direction up/down the slope. Dendrochronological samples were taken from the visibly healthy trees with no signs of damage in the trunk or crown. The sampled trees were dominant and co-dominant according to the Kraft classification [68]. The selection was examined randomly (RNG function, Excel) according to the distribution in the stand, which describes growth response (compared to sub-dominant and suppressed trees on each research plot [69]. The number of samples had to be sufficient for the EPS indicator described in Data Processing. A total of 104 samples were taken for dendrochronological analysis. The individual numbers of samples per area are described in the table of the site dendrochronology description of the research plots in the Results. Ring widths were measured to the nearest 0.01 mm with an Olympus binocular on a LINTAB measuring table and recorded with the TsapWin program [70].

### 2.3. Data Processing

2.3.1. Stand Structure and Analysis

The basic structure, diversity, and production characteristics of the tree layer were evaluated by the SIBYLA 5.1 software [71]. The input data were measured by individual dendrometric characteristics of trees (tree species, height, DBH, crown width, live crown base, and age), including coordinates. The PointPro 2.1 program (CZU, Prague, Czechia) was used to calculate the characteristics describing the horizontal structure [72] of tree individuals on the PRPs. The aggregation index was derived from all distances between the two nearest neighbors, the number of trees in the plot, plot area, and the perimeter of the plot [72]. The significance test of deviations from the values expected for a random arrangement of points was performed using Monte Carlo simulations. In the results, statistically significant values (exceeding the confidence interval) are marked with an asterisk. Next, structural diversity was evaluated based on the vertical Arten-profile index [73], diameter and height differentiation [74], crown differentiation, and total stand diversity (Table 3) [75]. The Arten-profile index was calculated using the basal area of tree species in individual stand layers [76]. Diameter and height differentiations are related to the ratio between the larger and the smaller diameter/height of all nearest neighboring trees [74]. The stand diversity index was calculated with regard to complex biodiversity [75]. Total diversity is composed of the following components of diversity: tree species diversity, diversity of vertical structure, diversity of tree spatial distribution, and diversity of crown differentiation. The input variables are the number of tree species, maximum and minimum tree species proportion, maximum and minimum tree height, maximum and minimum tree spacing, minimum height to crown base, and minimum and maximum crown diameter [77].

**Table 3.** The indices describing stand structure and their common interpretation.

| Criterion | Quantifiers | Label | Reference | Evaluation |
|---|---|---|---|---|
| Horizontal structure | Aggregation pattern | R (C&Ei) | [72] | mean value R = 1; aggregation R < 1; regularity R > 1 |
| Vertical structure | Arten-profile index | $A$ (Pri) | [73] | range 0–1; balanced vertical structure $A$ < 0.3; selection forest $A$ > 0.9 |
| | Vertical diversity | $S$ (J&Di) | [75] | low $S$ < 0.3, medium $S$ = 0.3–0.5, high $S$ = 0.5–0.7, very high $S$ > 0.7 |
| Structure differentiation | Diameter dif. <br> Height dif. | $TM_d$ (Fi) <br> $TM_h$ (Fi) | [74] | range 0–1; low $TM$ < 0.3; very high differentiation $TM$ > 0.7 |
| | Crown dif. | $K$ (J&Di) | [75] | low $K$ < 1.0, medium $K$ = 1.0–1.5, high $K$ = 1.5–2.0, very high $K$ > 2 |
| Complex diversity | Stand diversity | $B$ (J&Di) | [75] | monotonous structure $B$ < 4; uniform structure B = 4–5.9; non-uniform structure $B$ = 6–7.9; diverse structure B = 8–8.9; very diverse structure $B$ > 9 |

Notes: Monotonous structure = stands composed solely of a single tree species; vertically undifferentiated tree canopy; low variation in tree crown diameters; systematic spatial arrangement of trees. Uniform structure = stands composed of one to two tree species; vertical structure of the tree canopy formed by a single layer, occasional identification of a second layer; random horizontal structure of trees. Non-uniform structure = stands composed of up to four tree species with varied mixed proportions; vertical structure consisting of two to three tree layers; average crown size reaching 50%; random to weak clustering tree spatial pattern. Diverse structure = stands composed of an average of five canopy-forming tree species, with two to three having similar mixture proportions; irregularly moderately multilayered vertical structure, rarely differentiated; spatial arrangement of trees classified as heterogeneous with a tendency to cluster. Very diverse structure = forests characterized primarily by high biological diversity; vertically structured profiles forming multiple tree layers, containing up to seven canopy-forming tree species, of which at least three to four have relatively equal representation; highly varied crown widths; spatial arrangement of trees perceived as clustered [75].

The stand volume was calculated according to [78]. The relative stand density index (SDI) [79], the canopy closure (CC) [80], and the crown projection area (CPA) were observed for each PRP. The relative SDI was calculated as the ratio of the actual value of the stand density index to its maximum value. The stand density index represents the theoretical

number of trees per hectare if the mean quadratic diameter of the stand component is equal to 25 cm [79].

2.3.2. Dendrochronological Processing and Analysis

Dendrochronological data were analyzed in software R (version 4.3.1) [81] using the packages "dplr" [82,83] and "pointRes" [84]. Detrending of each measured sample was carried out by negative exponential detrending with a spline of 2/3 of the age of each tree using "dplr" instructions [85]. The detrended tree-ring growth data are averaged as ring-width index (RWI) that further describes site chronology for the research plot. An analysis of the pointer years through relative growth change was performed [86]. The pointer years reflect the number of standard deviations from the local mean of the average ring-width series in the previous four years. The pointer years identify event years where the pointer year > 0.75 standard deviation of the previous four years. The threshold of the percentage of trees in a negative or positive event year was used [87]. The pointer years and percentage mean annual growth deviation are distinguished by the most common event in the year class [84].

An expressed population signal (EPS) was carried out for the detrended data series. The EPS represents the reliability of a chronology as a fraction of the joint variance of the theoretical infinite tree population. The EPS was employed to represent the limit for using the dendrochronological data series concerning the climatic data. The significant EPS threshold for using the dendrochronological data is EPS > 0.85 [85]. The signal-to-noise ratio (SNR) indicates chronological signal strength. The SNR is a statistical metric that evaluates the strength of the targeted signal within a dataset of the series compared to the background noise level. A higher SNR value indicates a more robust climatic signal relative to noise. Inter-series correlations (R-bar) were calculated for the dendrochronological data series. The R-bar quantifies the similarity of tree-ring patterns among various samples. It represents the average pairwise correlation coefficient between individual trees within a chronology. A higher R-bar value signifies increased coherence among the tree-ring patterns [88]. First-order autocorrelation (Ar1) was also carried out. The Ar1 describes the degree of cross-correlation between a data point and the preceding one in a time series of tree-ring series. The EPS, SNR, R-bar, and Ar1 were calculated by the instructions to "dplr" [85] based on common dendrochronological theories [88,89].

2.3.3. Tree Rings and Climatic Analysis

The average tree-ring series of Turkey oak from research plots IT_1, BG_2, SK_3, and CZ_4 was correlated with climate data, namely precipitation and temperatures; 1968–2022 from weather stations in Italy (Potenza—720 m a.s.l.), Bulgaria (Divlya—600 m a.s.l.), Slovakia (Hurbanovo—115 m a.s.l.), and Czechia (Lednice—177 m a.s.l.) according to individual months and years. The DendroClim 2002 software [90] was used to model the radial growth depending on the climatic characteristics.

Spectral analyses of the data were performed using Statistica 13 software. The calculation was accomplished using the "Single Fourier (Spectral) Analysis" function, utilizing the "Periodogram" plot by "Period" output. The sine and cosine functions are mutually independent, also known as orthogonal. Therefore, we can aggregate the squared coefficients for each frequency to create the periodogram. The values in the periodogram can be understood in relation to the variance, representing the sums of the squares and of the data at the corresponding frequency or period [91]. While our datasets are in yearly intervals, the "period" in spectral analysis describes the length of the yearly interval cycles. The intensity of the cycles of our datasets indicates "periodogram values", which are expressed as the density of cycles per observation. This allows the identification of dominant frequencies or periods (cycles) in the data.

Seasonal temperature was determined by calculating the arithmetic mean of the monthly values within these seasonal months. For the calculation of seasonal precipitation, the sum of monthly precipitation totals during the respective seasonal periods was used.

The intentional selection of this seasonal window was intended to reduce variability at the start and end of the growing season. Thus, the seasonal data assessed within this timeframe accurately represent the shared vegetation period across all research plots.

A principal component analysis (PCA) was performed in the CANOCO 5 program [92] to evaluate the relationships between the stand structure, production parameters, radial growth, and research plots. This tool was used to reduce the dimensionality of a dataset while preserving the most important patterns, information, or relationships between the variables [93]. Prior to analysis, the data were standardized, centralized, and logarithmized. The results of PCA were presented in the forms of species and environmental variables ordination diagram. The input data to the PCA included the following parameters: stand volume, stem volume, basal area, diameter, slenderness coefficient, height, tree density, total diversity, diameter structure, vertical structure, and horizontal structure indices. The total number of variations was 48 (samples × species).

## 3. Results

### 3.1. Stand Characteristics, Production, and Diversity

Dendrochronological characteristics of the research plots in Table 4 reveal that all data exhibit significant EPS values for climate comparison with the ring-width index (RWI) of Turkey oak, indicating an EPS higher than 0.85 across the entire examined sample period. The sample count (No. trees) per plot varied from 24 to 29, with IT_1 = 24, BG_2 = 25, SK_3 = 26, and CZ_4 = 29 sample units. The mean ring width (RW) ranges from 2.09 to 3.13 across individual plots, with BG_2 having the lowest RW at 2.09 mm and SK_3 the highest at 3.13 mm. The Ar1 indicator suggests that research plots in BG_2 and IT_1 (Ar1 = 0.79; 0.60) exhibit higher to moderate levels of autocorrelation, indicating a substantial correlation between values in one year and those in the previous year. In contrast, plots in CZ_4 and SK_3 have (Ar1 = 0.41; 0.57), indicating a moderate to slightly lower degree of autocorrelation.

**Table 4.** Dendrochronological characteristics of Turkey oak stands on permanent research plots.

| PRP | No. Trees | Mean RW (mm) | SD RW (mm) | Mean Min–Max (mm) | Age Min–Max | Ar1 | R-Bar | EPS | SNR |
|---|---|---|---|---|---|---|---|---|---|
| IT_1 | 24 | 2.69 | 1.03 | 1.66–4.65 | 36–66 | 0.60 | 0.34 | 0.90 | 8.71 |
| BG_2 | 25 | 2.09 | 1.11 | 1.46–2.97 | 35–55 | 0.79 | 0.43 | 0.93 | 12.87 |
| SK_3 | 26 | 3.13 | 1.45 | 0.95–5.31 | 33–69 | 0.57 | 0.46 | 0.90 | 8.89 |
| CZ_4 | 29 | 2.23 | 1.06 | 1.60–3.09 | 40–63 | 0.41 | 0.55 | 0.97 | 29.60 |

Notes: No. trees—number of used core samples, mean RW—mean ring width, SD RW—standard deviation of ring width, mean min–max—mean ring-width range from the smallest to biggest tree, Age min–max—age range of the youngest and oldest sample tree, Ar1—first order autocorrelation, R-bar—inter-series correlation, EPS—expressed population signal, SNR—signal-to-noise ratio.

The number of live Turkey oak trees from 2021 to 2022 ranged between 475 and 775 trees per ha with an SDI of 0.52–0.82 (Table 5). The highest mean DBH (30.5 cm) and tree volume (0.708 m$^3$) was from inventoried stands in Italy, while the lowest values were observed in Bulgaria (19.0 cm, 0.182 m$^3$). In general, as tree density increases, tree dendrometric parameters decrease. The basal area ranged from 22.0 (BG_2) to 41.6 m$^2$ ha$^{-1}$ (IT_1), and the stand volume was from 141 m$^3$ ha$^{-1}$ (BG_2) to 407 m$^3$ ha$^{-1}$ (IT_1). The mean annual increment was from 2.82 m$^3$ ha$^{-1}$ in Bulgaria to 6.78 m$^3$ ha$^{-1}$ in Italy.

In terms of the horizontal structure, the spatial pattern of trees was prevailingly random or clustered in Bulgaria (Table 6). The vertical structure was quite variable (A 0.266–0.530), i.e., balanced on IT_1 to moderately differentiated on other PRPs. The diameter and differentiation index varied and reached low values, with the highest variability in Italy. Concerning the overall stand diversity, IT_1, BG_2, and SK_3 showed a uniform structure (B 4.507–5.075), whereas CZ_4 showed a monotonous structure (B 3.917). Simi-

larly, the lowest crown differentiation was observed in the Czech PRP compared to high crown diversity in Bulgarian stands.

**Table 5.** Overview of stand parameters of Turkey oak coppices.

| PRP | DBH | h | f | v | N | G | V | hd | MAI | CC | SDI |
|---|---|---|---|---|---|---|---|---|---|---|---|
| | (cm) | (m) | | (m³) | (tree ha⁻¹) | (m² ha⁻¹) | (m³ ha⁻¹) | | (m³ ha⁻¹ year⁻¹) | (%) | |
| IT_1 | 30.5 | 22.33 | 0.434 | 0.708 | 575 | 41.6 | 407 | 0.732 | 6.78 | 88.6 | 0.82 |
| BG_2 | 19.0 | 16.22 | 0.395 | 0.182 | 775 | 22.0 | 141 | 0.854 | 2.82 | 77.1 | 0.52 |
| SK_3 | 28.8 | 20.92 | 0.425 | 0.579 | 475 | 30.9 | 275 | 0.726 | 4.58 | 80.8 | 0.62 |
| CZ_4 | 28.2 | 21.80 | 0.402 | 0.547 | 625 | 39.1 | 342 | 0.773 | 5.70 | 87.4 | 0.79 |

Notes: DBH—mean quadratic diameter at breast height, h—mean height, f—form factor, v—average tree volume, N—number of trees, G—basal area, V—stand volume, hd—slenderness ratio, MAI—mean annual increment, CC—canopy closure, SDI—stand density index.

**Table 6.** Biodiversity of stands with Turkey oak on all PRPs.

| PRP | R (C&Ei) | A (Pri) | | S (J&Di) | | TM$_d$ (Fi) | | TM$_h$ (Fi) | | K (J&Di) | | B (J&Di) | |
|---|---|---|---|---|---|---|---|---|---|---|---|---|---|
| IT_1 | 0.896 | 0.266 | ↘↘ | 0.783 | ↗ | 0.314 | ↘ | 0.156 | ↘↘ | 1.219 | → | 4.507 | ↘ |
| BG_2 | 0.676 * | 0.385 | ↘ | 0.594 | → | 0.252 | ↘↘ | 0.147 | ↘↘ | 1.761 | ↗ | 5.075 | ↘ |
| SK_3 | 0.931 | 0.530 | → | 0.758 | ↗ | 0.220 | ↘↘ | 0.188 | ↘↘ | 1.490 | → | 4.693 | ↘ |
| CZ_4 | 1.206 | 0.426 | ↘ | 0.304 | ↘ | 0.163 | ↘↘ | 0.082 | ↘↘ | 0.714 | ↘ | 2.105 | ↘↘ |

Notes: R—aggregation index, A—Arten-profile index TM$_d$—diameter differentiation index, TM$_h$—height differentiation index, S—vertical diversity index, K—crown differentiation index, B—stand variability index statistically significant ($p < 0.05$) for horizontal structure (A—aggregation, R—regularity); arrows: ↘↘—low, ↘—low-medium, →—medium, ↗—high. * statistically significant aggregation spatial pattern ($\alpha = 0.05$) for horizontal structure (R index).

The diameter structure shown in Figure 2 demonstrates greater diameter variability with a broader distribution on the diameter spectrum in the case of Italy. The oak stands resembled the Gaussian curve in shape (typical of the same-age stand), which was the least flattened in the case of Bulgaria. Overall, oak was the most common in diameter classes, ranging from 18–23 (Bulgaria) to 28–33 cm (Czechia).

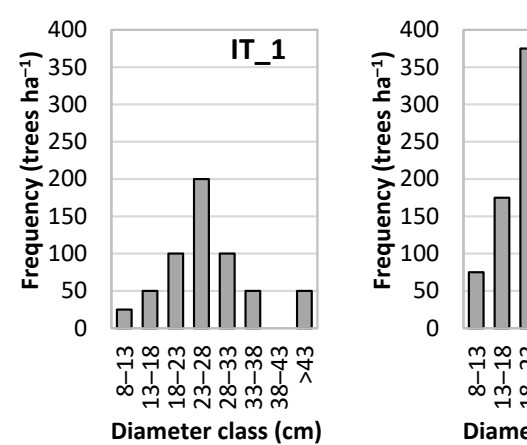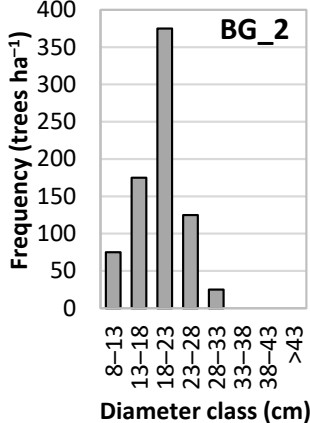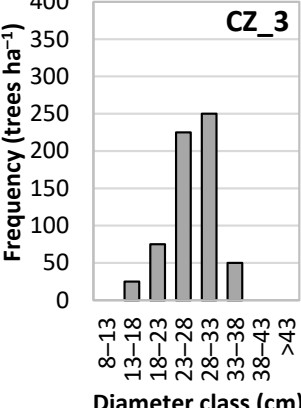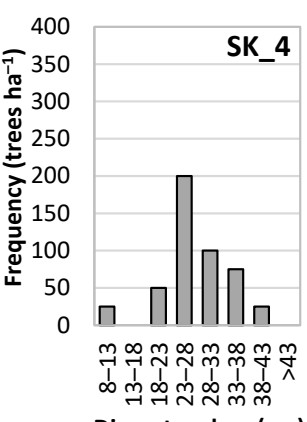

**Figure 2.** Diameter structure of oak forest stands according to countries.

*3.2. Tree-Ring Growth*

The ring-width increment of Turkey oak in the PRPs is different for each research plot (Figure 3). In terms of size, none of the PRPs have an increment lower than 0.5 or higher than 1.7 in RWI values. All research plots show irregular growth from one another. Overall, the tree-ring chronologies of Turkey oak were not subject to significant long-term fluctuations in the ring-width index over the study period. However, in the short term, the Central European PRPs fluctuate more in growth, as seen in the more irregular growth in CZ_4 and SK_3 from year to year. In contrast, the RWI of the PRPs in Southern Europe—IT and BG—show higher growth stability over the study period.

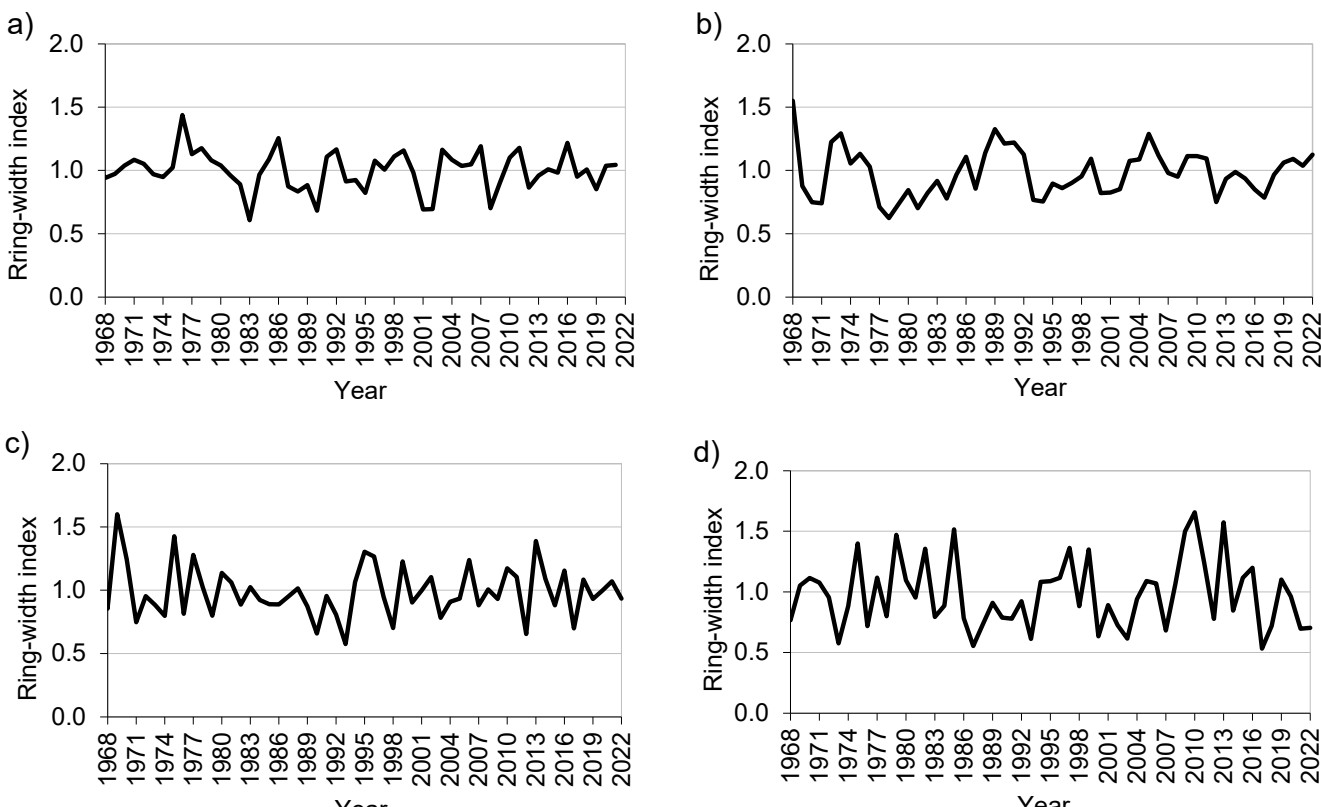

**Figure 3.** Standardized ring-width index chronology of Turkey oak in the period 1968–2022: (**a**) IT_1 (Italy); (**b**) BG_2 (Bulgaria); (**c**) SK_3 (Slovakia); (**d**) CZ_4 (Czechia).

The research plots in SK_3 and CZ_4 demonstrate a more fluctuating growth pattern in terms of annual oscillations of RWI from year to year. The annual RWI reveals that, in the case of SK_3, the RWI ranged from 0.65 in 2012 to 1.38 in 2013. Similar variability was observed in CZ_4, with notable fluctuations in the RWI, such as in 1978 (RWI = 0.80) to 1979 (RWI = 1.47). Significantly smaller RWI fluctuations were observed in plots in IT_1 and BG_2. This variation in plots from southern regions of Europe exhibits considerably smaller annual differences. For instance, from 1983 to 1984, IT_1 experienced a higher RWI increase, but only from 0.60 to 0.96. Similar small fluctuations are seen in BG_2, where notable RWI increases occurred, for example, in 1987 to 1988, with the RWI rising from 0.86 to 1.13.

Mean growth deviations in RWI are described in Figure 4 for research plots in IT_1, BG_2, SK_3, and CZ_4. Two climatically significant years were recorded for BG_2 in 1979 and 1980. In CZ_4, the negative climatic years were 1973 and 2017. From the viewpoint of negative climatic years, there is no noticeable difference between the PRPs in Southern and Central Europe.

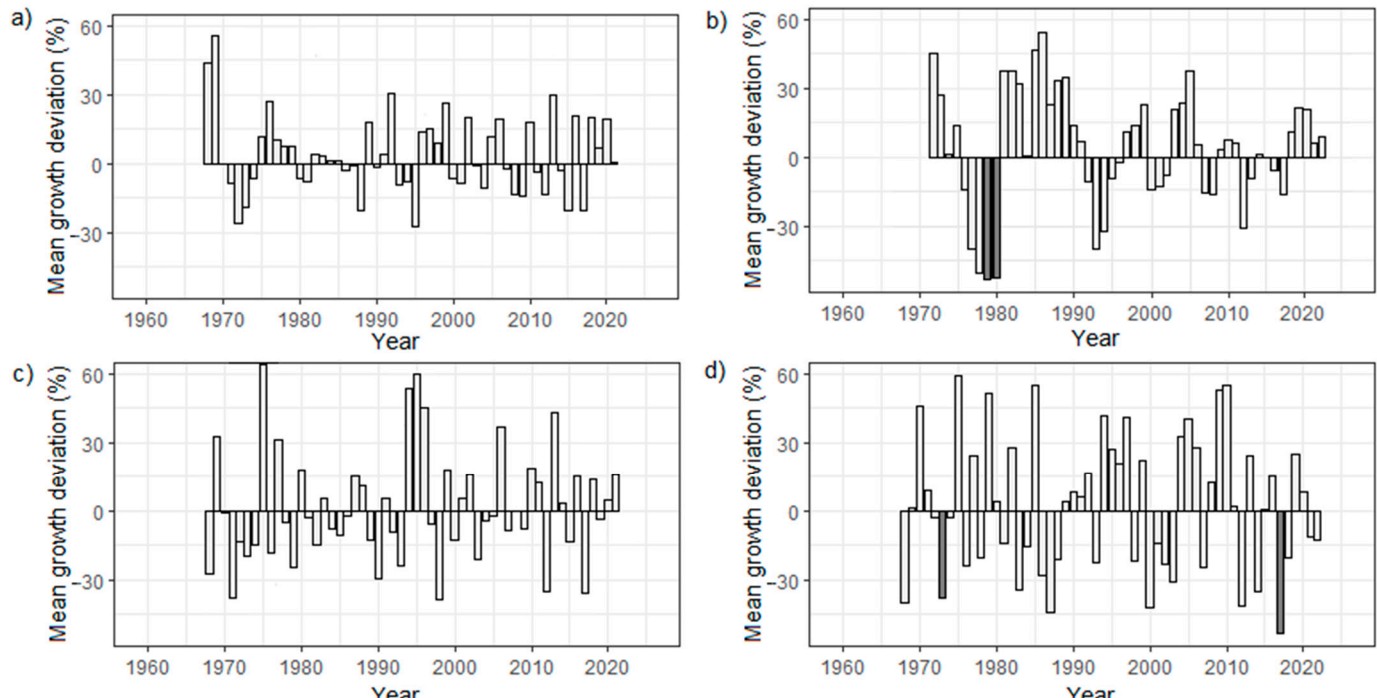

**Figure 4.** Percentage of mean growth deviation of Turkey oak for the period 1960–2022 with the pointer years (highlighted in dark grey); (**a**) IT_1 (Italy); (**b**) BG_2 (Bulgaria); (**c**) SK_3 (Slovakia); (**d**) CZ_4 (Czechia).

In terms of mean deviations and their oscillations, it is evident that plots in the southern regions of Europe—IT_1 and BG_2—exhibit a smoother transition between negative decline and growth deviations. In contrast, plots in Central Europe—SK_3 and CZ_4—show more regular oscillations in the transition from negative to positive, with these values oscillating mainly from year to year.

### 3.3. Turkey Oak's RWI with Monthly Precipitation and Temperature

The correlation coefficients in Figure 5 show the course of RWI against the monthly averages of temperatures and precipitation in the research plots. The radial growth of oak in sites in Italy and Bulgaria was least affected by monthly temperature and precipitation development (two significant months). Contrarily, in plots of Central Europe (SK_3 and CZ_4), the climatic factors studied had a significant influence on increment development (four significant months). Overall, the radial growth of Turkey oak correlates significantly ($p < 0.05$) positively with the course of monthly precipitation and negatively with the course of monthly temperatures. Unlike BG and IT, Central European PRPs are significantly more correlated with precipitation when compared to temperature. In comparison, precipitation and temperature for the Italian and Bulgarian PRPs are equally weighted. Generally, temperatures from June to August indicate a negative effect on growth, and precipitation correlates from March to June. These are the primary limiting factors for the radial growth of Turkey oak. Overall, temperatures exert a predominately negative influence on the RWI across all areas during the season, while precipitation has a significantly more positive impact on the RWI.

The results also indicate that precipitation is more frequently correlated in Central Europe on plots in SK_3 and CZ_4. In contrast, fewer correlations with lower values are observed in plots from the southern parts of Europe in IT_1 and BG_2.

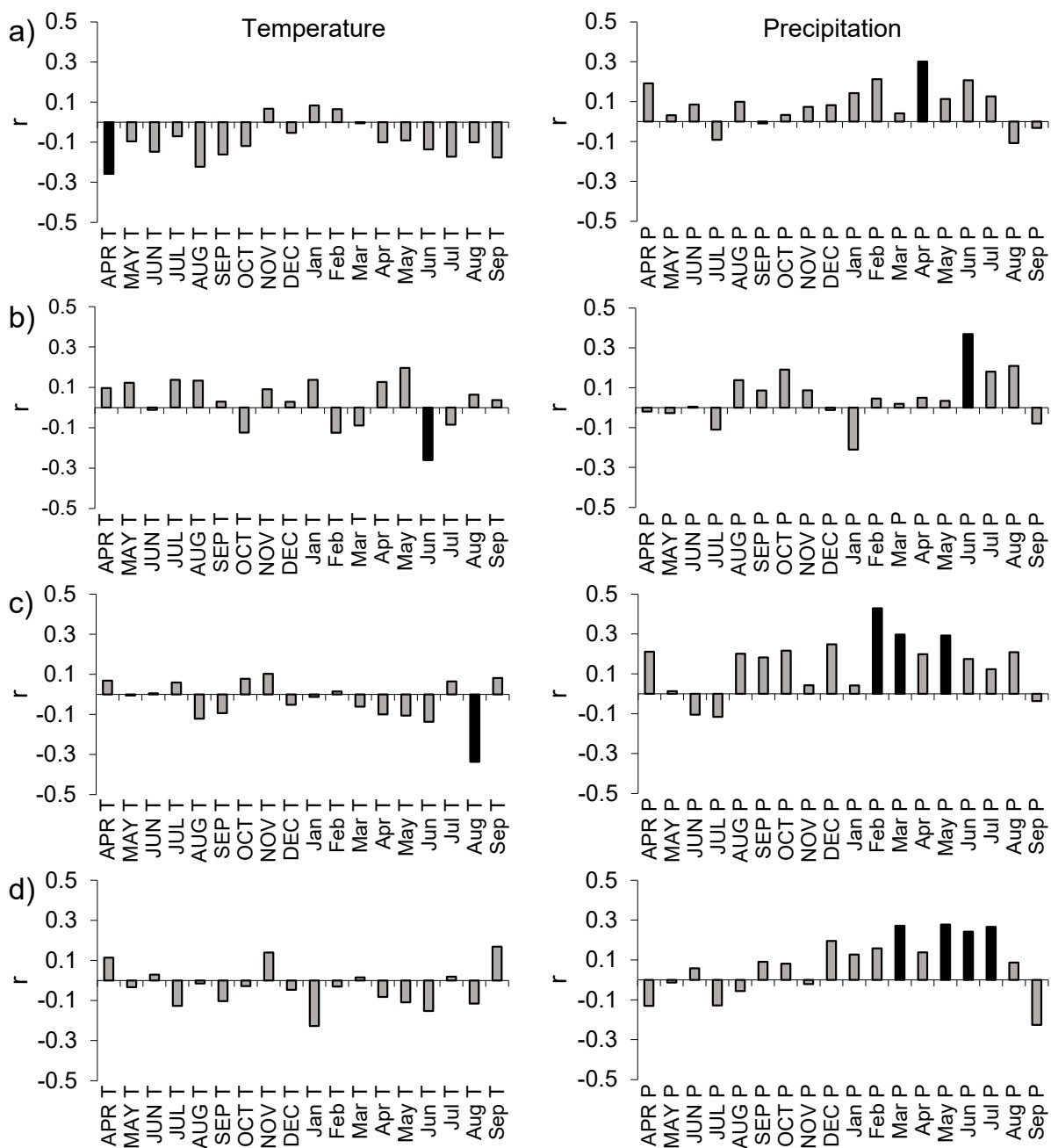

**Figure 5.** Correlation coefficients of the site chronology of the Turkey oak ring-width index with the average monthly air temperature (left) and sum of precipitation (right) from April to December of the previous year (uppercase letters) and from January to September of the current year (lowercase letters) in the relative year derived from the period 1968–2022: (**a**) IT_1 (Italy); (**b**) BG_2 (Bulgaria); (**c**) SK_3 (Slovakia); (**d**) CZ_4 (Czechia). Significant months ($p < 0.05$) are highlighted in black.

*3.4. Turkey Oak's RWI Growth Cycles*

Spectral analysis shows that the study plots in Italy (IT_1) and Bulgaria (BG_2) undergo longer 6 to 8-year growth cycles in RWI cycles (Figure 6). Research plot IT_1 shows the longest cycles with a frequency of 6.2 years. Research plot BG_2 illustrates the longest growth cycles in RWI with a frequency of 7.7 years. Contrastingly, the research plots in Central Europe (SK_3 and CZ_4) experience significantly shorter cycles ranging from 2.4 to 4.8 years. The RWI growth cycles of Turkey oak in the Central European sites are shorter than in Southern Europe, so it can be determined that Central Europe has the most frequent RWI cycles of 2 to 5 years. Southern European sites of Turkey oak have more pronounced

6 to 8-year RWI cycles. Overall, on the more northerly plots in SK_3 and CZ_4, the growth cycles of RWI are shorter and much more intense, as indicated by spectral analysis, which reveals predominantly 2 to 5-year cycles. In contrast, in Southern Europe research plots, the growth cycles of RWI are longer and much more intense over a longer time span, ranging from 6 to 7 years.

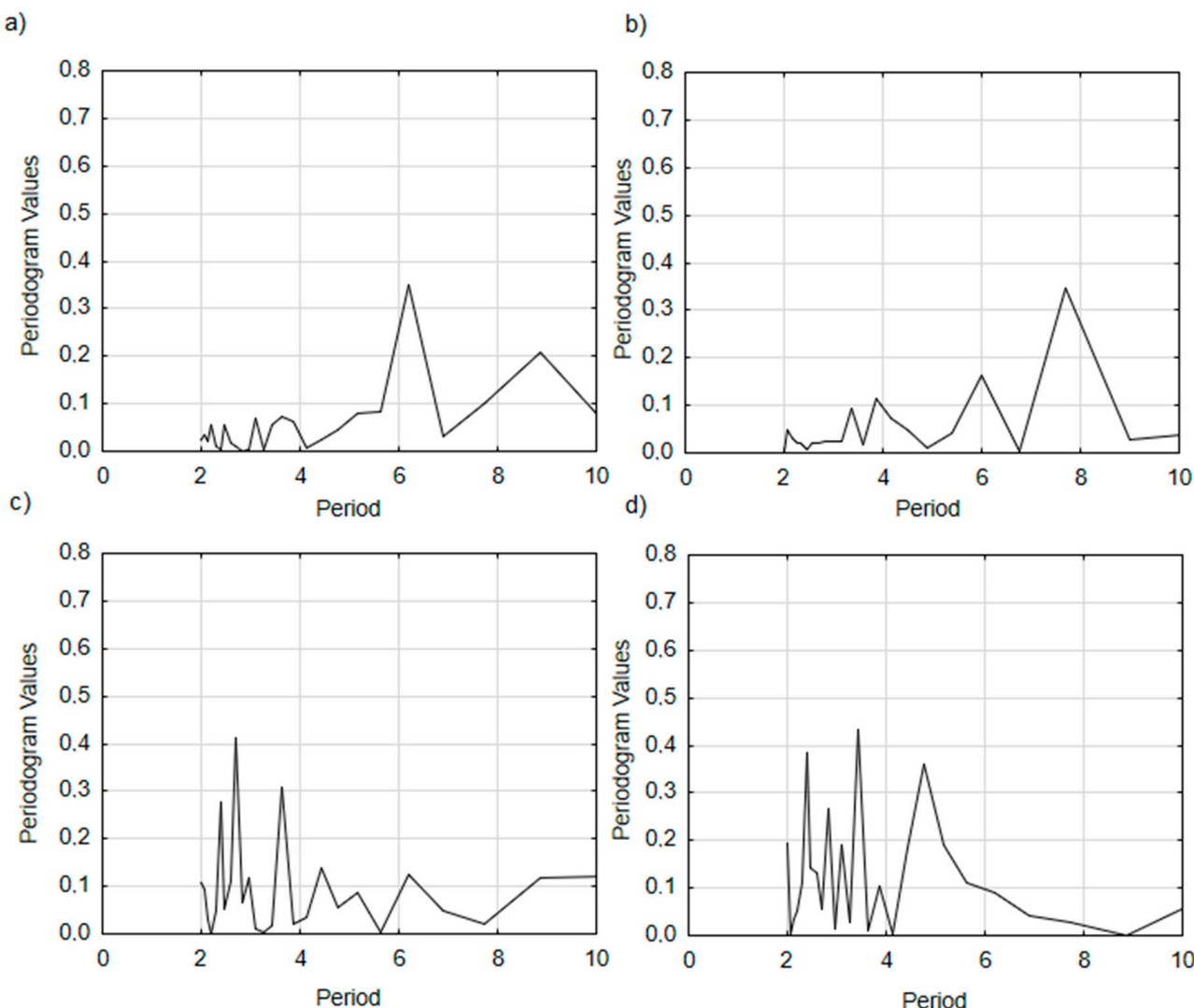

**Figure 6.** Single spectral analysis of ring-width index (RWI) of Turkey oak from 1968 to 2022: (**a**) IT_1 (Italy); (**b**) BG_2 (Bulgaria); (**c**) SK_3 (Slovakia); (**d**) CZ_4 (Czechia).

*3.5. Interaction between Stand Structure, Production Parameters, and Growth*

The results of PCA are presented in an ordination diagram in Figure 7. The first ordination axis explains 62.2% of data variability, and the first two axes together explain 83.6%. The x-axis illustrates the stand volume and basal area, and the y-axis represents the vertical structure (A index) combined with the diameter structure (TM$_d$ index). The total diversity was positively correlated with diameter differentiation, while these indices were negatively correlated with horizontal structure (tendency to regular spatial pattern). Production parameters, such as tree height, stand volume, basal area, DBH, and tree volume, were positively correlated to each other. The radial growth of oak increased with a decreasing number of trees in stands and the slenderness coefficient. The vertical structure (A index) was the lowest explanatory variable in the ordination diagram.

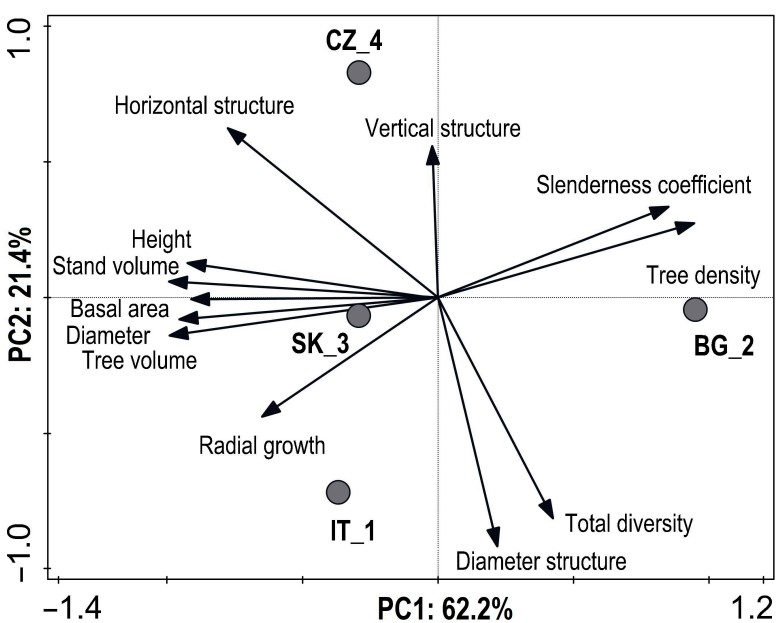

**Figure 7.** Ordination diagram showing the results of the PCA analysis of relationships between stand characteristics (Stand volume, Stem volume, Basal area, Diameter, Slenderness coefficient, Height, Tree density), diversity (Total diversity—B index, Diameter structure—$TM_d$ index, Vertical structure—A index, Horizontal structure—R index), and radial growth. Grey symbols indicate research plots.

## 4. Discussion

### 4.1. Production Potential and Stand Density

Regarding basic production parameters, the average number of trees in the study plots ranged from 475 to 775 trees $ha^{-1}$, the stand basal area ranged from 22 to 42 $m^2$ $ha^{-1}$, and the volume of roundwood (timber to the top of 7 cm o.b.) ranged from 141 to 407 $m^3$ $ha^{-1}$. With regard to the distribution of Turkey oak, most of the comparable data on similar stands came from Italy. For example, stand characteristics of Turkey oak coppices at 55 years of age in the Italian Mediterranean region corresponded to an average observed number of 578 to 1018 trees $ha^{-1}$, with a stand basal area ranging from 29 to 35 $m^2$ $ha^{-1}$, and the stand volume was similar, ranging from 260 to 308 $m^3$ $ha^{-1}$ [94]. Another study from Central Italy reported that in stands of over-aged coppices dominated by Turkey oak at the age of 55 years, the mean number of trees was higher again, ranging from 794 to 891 trees $ha^{-1}$, and the stand basal area reached values of 29 to 30 $m^2$ $ha^{-1}$, while the stand volume reached lower values in a very narrow range of 230 to 231 $m^3$ $ha^{-1}$ [95]. Šrámek et al. [96] cite that the number of trees in over-aged coppices in Türkiye aged 70–75 years was around 577 trees $ha^{-1}$, and the stand basal area reached 38 $m^2$ $ha^{-1}$. However, other tree species such as sessile oak (*Quercus petraea*), Italian oak (*Quercus frainetto*), European hornbeam (*Carpinus betulus*), or sweet chestnut (*Castanea sativa*) were also significantly represented in these Turkish oak stands. In this case, it is necessary to consider the species composition of the tree layer, which significantly influences the production characteristics of the coppice forest. A significantly higher number of trees (1870 trees $ha^{-1}$) in oak coppice stands at 71 years of age was reported by [97]. Again, the significant number of trees is a result of the diverse species composition of the coppice forest, with the presence of not only oaks but also small-leaved lime (*Tilia cordata*) or European hornbeam (*Carpinus betulus*).

With regard to stand density, the number of oak trees was distinctly correlated with tree-ring growth. The radial growth of oak increased with a decreasing number of trees in stands and a closely related slenderness coefficient. Similarly, mean tree growth was significantly higher under low density compared with maximum stand density in the case of sessile oak (*Quercus petraea* [Matt.] Liebl.) [98]. A significant response of radial growth

to different stand densities was also confirmed for other tree species [99]. The effects of competition on tree radial growth were found to be much higher for shade-intolerant species, such as oaks [100]. Moreover, the reduction in tree density increases subsequent growth for remaining trees and decreases sensitivity to climate change, especially drought stress [101–103]. The resilience of trees after reducing the stand density was confirmed, e.g., for Scots pine stands due to the lower competition of remaining trees to available water sources with growing resistance during a drought event [104].

### 4.2. Stand Biodiversity and Structure

Stand diversity (and structure) significantly influences growth processes and tree resilience to climate change [4,105]. Šimůnek et al. [66] describe the disparate response of radial tree growth in homogeneous vs. heterogeneous stands to climatic factors. Therefore, a detailed analysis of the structure and diversity of the studied stands is an essential starting point for further dendrochronological analyses. In the case of the evaluated stands of Turkey oak coppice forests aged 50 to 60 years, the overall stand diversity was very low, or rather, the stands reached a monotonous to uniform structure (B index 3.92–5.08). A significantly higher overall diversity (B index 9.73–10.46) of oak coppice forests in Czechia is described by [97], chiefly due to the addition of other broadleaved species and leaving the stands to develop spontaneously. In terms of the horizontal structure of the tree layer, the spatial pattern of the studied stand was predominantly random, with a tendency to aggregate or cluster in Bulgaria. Another study from Czechia describes a similarly aggregated structure of oak coppice forests, especially in the initial distances between trees of up to 2 m [39]. Of all the parameters studied, crown differentiation reached the highest values of structural diversity [39].

### 4.3. Turkey Oak Coppices and Climate

Radial growth of Turkey oak research plots in this study were clearly more influenced by climatic factors (monthly temperatures and precipitation) in Central European countries than in Italy and Bulgaria. Overall, RWI growth significantly correlated positively to the course of monthly precipitation ($p < 0.05$), especially from March to June, and negatively to the course of monthly temperatures, especially from June to August. Amorini et al. [106] show minimal increments in years with substantial drought on Turkey oak in Central Italy. In addition to the positive effect of May and June precipitation on radial growth, this study also shows a positive correlation for May and June minimum temperatures and the March and April maximum temperatures of the current year. In contrast, no statistically significant correlation was found between the previous year's climate data and radial growth. Similar results were also found in a study by [107] carried out in Turkey oak stands in Slovakia, where it was confirmed that the most significant positive effect on the magnitude of growth is due to the amount of precipitation during the growing season, especially in May and June. Another study from Central Italy confirms the importance of the May and June rainfall of the current year [95]. Conversely, the effect of temperature on the radial growth of Turkey oak is not as significant as the effect of precipitation, as in the case of this study or the study in Bulgaria [108].

In Central Europe, specifically in the plots SK_3 and CZ_4, a higher number of observed positive correlations between RWI and precipitation were identified when compared to the Southern regions represented by IT_1 and BG_2. In the research plots of Central Europe, seven significant correlation coefficients were found, while only three significant correlation coefficients were observed in Southern Europe. For instance, in Northern Europe, the English oak (*Quercus robur* L.) exhibits strong positive associations with precipitation, particularly during the summer, where growth anomalies are linked to oak growth, especially in combination with higher temperatures [109]. The lower positive correlations with precipitation are linked to the lower available precipitation during the vegetation season in Southern Europe, while this general statement is also well documented [110].

*4.4. Tree-Ring Growth Cycles across the Studied Plots*

The Turkey oak has an advantage over other Central European oaks in that it is known to be significantly more drought tolerant, but at the same time, its growth is not as aggressive as, for example, sessile oak or English oak (*Quercus robur* L.) [64,111,112]. The results in the investigated plots in Southern Europe show that Turkey oak grows steadily, and the largest fluctuations in growth are recorded specifically concerning monthly temperatures and lack of precipitation during the summer months. This is accompanied by the results of the spectral analyses, which show that the tree-ring chronologies of Turkey oak in Italy and Bulgaria fluctuate between six and eight years. These cycles of around seven years are most often associated with the 7-year temperature cycle, which is typical of the European continent [113,114]. The tree-ring series of Turkey oak may be most closely associated with the temperature cycle. Contrastingly, the study plots in Central Europe (SK_3 and CZ_4) follow a 2 to 5-year cycle, which, again, corresponds to a significantly shorter precipitation cycle in Europe [115–117]. Based on these findings, a greater influence of precipitation on tree-ring increment is evident in Central Europe research plots, as indicated by the stronger correlations. Thus, the drier climate of Southern Europe produces a longer cycle of tree-ring increment compared to plots located in Central Europe. The cycles studied may also be closely related to fructification. Masting of Turkey oak in Southern European countries is around five to seven years but can be shorter based on sufficient moisture [118,119]. This correlates with the results found in this study or with the tree-ring series from Italy and Bulgaria (6 to 8-year cycle). On the contrary, as mentioned, in Central Europe, due to abundant precipitation, the situation is different and seed years repeat significantly more frequently in 2 to 5-year cycles. It is due to higher precipitation during the growing season in Central Europe, where acorn fruiting occurs more frequently in Turkey oak, which is close to the natural 2-year period [120]. Tree seed production is associated with tree-ring growth and the influence of weather in both the previous and current growing seasons [121–123]. The frequency of tree-ring growth can serve as an indicator of the theoretical fertility of trees, where we observed 2 to 5-year cycles.

*4.5. Potential for Coppice Forests*

Due to climate change, rising temperatures, and more frequent long-term droughts, the species composition and stand structure will change in the coming decades [124]. The Turkey oak may become a valuable alternative tree species with great potential for adaptation to changing environmental conditions, especially in Central Europe [4]. This tree species generally needs lower amounts of air and soil moisture for its growth [125]. The resilience of Turkey oak to climatic extremes is also confirmed by [112], comparing the effect of the climate on the growth of Turkey oak and sessile oak in northern Hungary, proving that Turkey oak can better recover from prolonged periods of drought. Similar characteristics are described in this study, indicating the high resilience of this tree species, which is documented by the low number of negative years in the RWI and the relatively small influence of climatic factors. This is because after the second rotation of the coppice, surface roots are being formed as the old ones are already dying [108]. Generally, high temperatures in the growing season can induce increased water stress and a subsequent reduction in radial growth due to increased water loss through evapotranspiration and soil moisture evaporation. Stafasani & Toromani [126] reported that most of the coppice Turkey oak mixture stands in Albania showed extreme drought in June of the current year as a limiting factor for growth. The negative effect of June and July temperatures on the growth of young oaks has previously been observed at several sites in continental Europe [127,128] in northern Spain [129] and also in the Mediterranean [130,131]. Moreover, the positive relationship of radial growth with June precipitation shows that water balance in this month is critical for phenology [132,133].

### 4.6. Study Limitations and Ideas

The number of samples collected for dendrochronological analysis was sufficient, which was confirmed by EPS analysis. The gathered data exhibit satisfactory EPS ranging from 0.90 to 0.96. It is crucial to note that the minimum EPS threshold for data utilization is 0.85, as stipulated by this indicator [88,134]. Furthermore, the collected data, supported by EPS, demonstrate that an ample number of samples has been collected to describe the chronology relating to climate, indicating a robust data series [135].

Before the end of this discussion, it is necessary to mention that this study was constrained by the limited number of research plots in each of the evaluated countries. Consequently, the study was not focused on a detailed description of regional climatic conditions but rather on depicting conditions within the research plots. For this reason, information from the nearest meteorological stations in the studied locations was also utilized. The repetition of research plots is ensured by two robust chronologies from Central Europe and two from Southern Europe. This study presents information on tree-ring growth frequency in Central and Southern Europe.

On the other hand, similar future studies, in addition to the limitations mentioned above (such as the small number of plots), should also focus on other important factors influencing the structure, production, and response of trees to climate change, such as genetics. The provenance of tree species significantly affects the production potential of wood, carbon sequestration, and resistance to climatic extremes [136,137]. Significant genetic diversity was also found in Turkey oak [138,139]. In the future, further research should also focus on the influence of various silviculture regimes or the admixture effect of other tree species in oak stands in the context of adaptation to climate change.

### 5. Conclusions

In conclusion, it could be argued that coppice forests represent a suitable alternative to standard forest management practices. The suitability can be affected by tree species composition, especially during ongoing climate change. This was confirmed in the presented study where the evaluated Turkey oak coppice forests in Italy, Bulgaria, Slovakia, and Czechia showed, on the one hand, relatively high resistance to unfavorable climatic factors, including climatic extremes, and on the other, adequate values of timber production.

The lowest influence of climatic factors on growth was found in Italy and Bulgaria compared to the tree's climate sensitivity in Central Europe—on the northern edge of its natural distribution range. The spectral analysis also showed that the research areas in Southern Europe go through longer 6 to 8-year growth cycles in radial growth compared to Central Europe (shorter cycles of 2.4 to 4.8 years). It was found that the main limiting factor for growth was the lack of precipitation during the growing season, whereas temperatures played almost no role in the radial growth processes. Regarding the fact that in recent years there has been significant warming in Europe, therefore Turkey oak can be identified as a crucial tree species in terms of adaptation strategies to climate change. In general, this study is the basis for understanding and predicting the growth responses of Turkey oak coppice stands to the climate under conditions of ongoing global climate change in Europe. For future long-term research, however, it is necessary to further focus on other factors, such as genetic origin or different silviculture practices in the context of climate change.

**Author Contributions:** M.Š. and S.V. designed the research. M.Š., Z.V., V.H. and L.B. collected samples in the field. V.Š., V.H. and A.P. analyzed the data. I.Š., Z.S. and L.B. sourced meteorological data. M.Š., S.V., I.L., Z.V., V.Š. and J.C. prepared first draft of the manuscript, All authors have read and agreed to the published version of the manuscript.

**Funding:** This research was funded by the Internal Grant Agency, Faculty of Forestry and Wood Sciences, Czech University of Life Sciences Prague (IGA FFWS 2021, Project No A_21_17). This research received funding from the LIFE Climate Action sub-programme of the European Union—project CLIMAFORCEELIFE (LIFE19 CCA/SK/001276).

**Data Availability Statement:** Monthly temperature and precipitation data for the Czech Republic are available from the Czech Hydrometeorological Institute (www.chmi.cz, accessed on 25 January 2023). Monthly temperature and precipitation data for Slovakia are available from the Slovak Hydrometeorological Institute (www.shmu.sk, accessed on 25 January 2023). Monthly climatic data for Italy are available from the Italian Civil Protection Authority, Basilicata Region (www.centrofunzionalebasilicata.it, accessed on 16 February 2023). Monthly climatic data for Bulgaria are available from the National Institute of Meteorology and Hydrology (www.weather.bg, accessed on 16 February 2023). The tree-ring data presented in this study are available on request from the corresponding author.

**Acknowledgments:** Acknowledgement goes to the Czech Hydrometeorological Institute, Slovak Hydrometeorological Institute, Italian Civil Protection Authority, and National Institute of Meteorology and Hydrology in Bulgaria for providing the datasets. We would also like to thank both Richard Lee Manore, a native speaker, and Jitka Šišáková, an expert in the field, for checking the English of this paper.

**Conflicts of Interest:** The authors declare no conflict of interest.

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
