# Peer review of "Turkey Oak (Quercus cerris L.) Resilience to Climate Change: Insights from Coppice Forests in Southern and Central Europe"

_forests, doi:10.3390/f14122403_

Round 1

Reviewer 1 Report

Comments and Suggestions for Authors

Dear Authors,

I would like to congratulate you on the quality of your manuscript. Your research has the potential to make a significant contribution to the field of forest sciences and dendrochronology. However, I would like to direct your attention to some issues that can further enhance the article

There are eight questions throughout the text that deserve careful review. Ensure that each of them is addressed accurately. This will help strengthen the clarity of the article.

I am available for any further clarifications or to discuss any specific aspects of the questions in question.

Sincerely,

Reviewer

Author Response

Answer: We would like to sincerely thank the reviewer for the expert revision of the article. In this response, we would like to respond to some of the important comments that were raised on the PDF by reviewer.

Response to the length of the growing season: We have added a comment to the methodology that explains why we decided to use this time window. "Seasonal temperature was determined by calculating the arithmetic mean of the monthly values within these seasonal months. For the calculation of seasonal precipitation, the sum of monthly precipitation totals during the respective seasonal periods was used. The intentional selection of this seasonal window aimed to reduce variability at the start and end of the growing season. Thus, the seasonal data assessed within this timeframe accurately represent the shared vegetation period across all research plots." The column was removed from the table.

Response on table 4: It was moved to result section.

Response on number of samples: It was described for Table 4 in results section.

Response on spectral analysis methodology: Inserted.

Response on reduction in tree density reduces sensitivity to climate changes, especially to water stress: The paragraph was improved.

Reviewer 2 Report

Comments and Suggestions for Authors

Journal: Forests (ISSN 1999-4907)

Manuscript ID: forests-2692878

Title: Turkey Oak (Quercus cerris L.) Resilience to Climate Change: In-sights from Coppice Forests in Southern and Central Europe

Overall  Comments and Suggestions for Authors

To authors,

Regarding the stand structure and production potential to manage Turkey oak (Quercus cerris L.) stands, this manuscript might be interesting to the relevant researchers who deals with similar issues such as silviculture, coppices, growth and yield, and climatic responses of forests. Overall, the manuscript contents and structure were understandable, and the objectives of the study were clear. However, from my point of view, it is still considered that the number of sample plots and measurement instances were not sufficient to provide any research findings related to the topic. Because of this issue, the analytic results were mostly sample cases. Although I acknowledged the authors’ effort to collect the data, analyze the results, and prepare the manuscript, I evaluate that this manuscript may be not suitable to be published in this international journal, Forests. Authors can receive all the reviewer’s comments and Editor will provide the decision. At present, I gave my comments in detail as follows. I wish the results can be informative to the relevant researchers in this topic.

I hope that this manuscript can be improved based on peer-review’s comments. My specific comments were provided in detail as follows.

Kind regards,

Reviewer

Point 1.

I consider that Introduction and Materials and Methods were rather clear and structured well. However, one critical point is that the number of sample plots and the measurement instances. It was made based on four plots and there was no re-measurement. Instead, authors tried to collect the data using increment cores. However, the number of sites is basically limited to four sites. Therefore, the results handled in this study were just sample cases and it varied much as authors showed. I do not think that it gives us any convincing consistent research findings from sufficient observations.

Point 2.

In Results, to figure out the stand structure as stated in the objectives, the distribution and correlation of stand characteristics can be considered as interesting results. For example, authors would have analyzed the distribution of DBH or height. Also, some correlation with scatter plots between two major factors.

Point 3.

Working for the data of the increment cores is also arduous and valuable. Why don’t authors try to show the growth by tree characteristics, for example by crown class? Also, the results in relation to climatic factors would be interesting to show the diameter growth, not just with year on X-axis.

Point 4.

Regarding Figure 4, just one previous year is not enough to analyze and derive a conclusion. Again, due to the lack of sample plots, it just showed case studies. It cannot provide any convincing pattern by month. The amount of temperature or precipitation itself may be better to be used in the analyze directly with diameter growth.

Point 5.

Overall, due to the restricted number of sample plots, the results were not convincing enough as it showed no statistically significant pattern yet. Therefore, I am skeptical to proceed with this manuscript in the journal, Forests.

Author Response

To authors,

Regarding the stand structure and production potential to manage Turkey oak (Quercus cerris L.) stands, this manuscript might be interesting to the relevant researchers who deals with similar issues such as silviculture, coppices, growth and yield, and climatic responses of forests. Overall, the manuscript contents and structure were understandable, and the objectives of the study were clear. However, from my point of view, it is still considered that the number of sample plots and measurement instances were not sufficient to provide any research findings related to the topic. Because of this issue, the analytic results were mostly sample cases. Although I acknowledged the authors’ effort to collect the data, analyze the results, and prepare the manuscript, I evaluate that this manuscript may be not suitable to be published in this international journal, Forests. Authors can receive all the reviewer’s comments and Editor will provide the decision. At present, I gave my comments in detail as follows. I wish the results can be informative to the relevant researchers in this topic.

I hope that this manuscript can be improved based on peer-review’s comments. My specific comments were provided in detail as follows.

Kind regards,

Reviewer

Answer: As an opening word, we would like to sincerely thank the reviewer for his review and the time to rate our manuscript. Although the reviewer has important comments, we tried to respond to these comments as accurately and with the greatest care as possible. We edited the manuscript in such a way as to satisfy the reviewer as much as possible.

Point 1.

I consider that Introduction and Materials and Methods were rather clear and structured well. However, one critical point is that the number of sample plots and the measurement instances. It was made based on four plots and there was no re-measurement. Instead, authors tried to collect the data using increment cores. However, the number of sites is basically limited to four sites. Therefore, the results handled in this study were just sample cases and it varied much as authors showed. I do not think that it gives us any convincing consistent research findings from sufficient observations.

Answer: In response to this comment, we must clarify that we have the sufficient number of samples from which we have constructed a dendrochronological curve. This can be substantiated by the EPS agreements, which exceed 0.85, a threshold we meet. Furthermore, our data indicates that the measured dendrochronological data are consistent and correlate with each other, providing robust data for further analysis. According to dendrochronological theories, our data sets are suitable for comparing with climate data. Additionally, in our hypotheses, we are comparing growth differences between various research plots in different states.

On the other hand, we do not describe regional chronologies but rather site-specific chronologies for the Quercus cerris. From this perspective, we have modified the text to emphasize that it is a site-specific chronology. In terms of our methodology, we do not create a chronology for the entire state but rather for the specific research plot. Nevertheless, our data are comparable and describe the growth frequency, as evident from our results and in comparison with plots from other states.

To address potential inaccuracies in our methodology, we have edited the manuscript text to highlight the regional nature of the results. Despite this regional focus, the results remain comparable due to high EPS and the robust nature of dendrochronological outcomes like r-bar. It's crucial to mention that we have also added a discussion section addressing the advantages and limitations of our study, aiming to describe and formulate these discrepancies more effectively.

Point 2.

In Results, to figure out the stand structure as stated in the objectives, the distribution and correlation of stand characteristics can be considered as interesting results. For example, authors would have analyzed the distribution of DBH or height. Also, some correlation with scatter plots between two major factors.

Answer: We added graphs for DBH. These graphs better explain the structure for diameter in the research plots. We hope that new results will improve the overall picture for the structure part.

Point 3.

Working for the data of the increment cores is also arduous and valuable. Why don’t authors try to show the growth by tree characteristics, for example by crown class? Also, the results in relation to climatic factors would be interesting to show the diameter growth, not just with year on X-axis.

Answer: The creation of growth curve assessments in categorizing thickness classes of DBH cannot be evaluated solely based on the sample collection methodology. Samples were exclusively taken from overstory and canopy trees, as outlined in the methodology „from the dominant and co-dominant trees according to the Kraft classification“. These upper canopy trees, in good health, serve as better climatic indicators, thereby enhancing the overall site chronology's quality. The upper and top canopy of the tree layer carries more weight for climatic analyses. Subcanopy and shadowed trees have a low to zero value for comparing tree rings with climate. Additionally, a substantial amount of signal noise in the chronology is present in subcanopy shadowed trees, leading to low mutual correlations among samples in reality. This would prevent us from establishing a robust chronology for such tree classes.

Point 4.

Regarding Figure 4, just one previous year is not enough to analyze and derive a conclusion. Again, due to the lack of sample plots, it just showed case studies. It cannot provide any convincing pattern by month. The amount of temperature or precipitation itself may be better to be used in the analyze directly with diameter growth.

Answer: We tried to respond to this question in Point 1 and in the discussion chapter of study limitations. However, we believe that, despite the fact that we do not have a large number of surveyed research plots, we have sufficiently robust data for our conclusions. Therefore, we have robust international datasets that deserve to be published.

Point 5.

Overall, due to the restricted number of sample plots, the results were not convincing enough as it showed no statistically significant pattern yet. Therefore, I am skeptical to proceed with this manuscript in the journal, Forests.

Answer: We are arguing that we have a sufficient number of mutually similar dendrochonological samples for the analysis of reactions to climate. Even one plot with a significant EPS is adequate, as it describes a robust site chronology. Given the nature of our results, the cyclical growth frequencies will be similar across the entire state, as this involves analyzing the climate's impact on tree-ring series. Furthermore, the results themselves support a higher frequency of precipitation in Central Europe compared to the south. This single result alone is a significant indicator that coppices respond to climate and it deserves to be published.

Reviewer 3 Report

Comments and Suggestions for Authors

The paper entitled “Turkey Oak (Quercus cerris L.) Resilience to Climate Change: Insights from Coppice Forests in Southern and Central Europe” reflects the development of applied research on the topic of coppice growth and stand structure diversity. The manuscript needs to be improved. Some issues in the introduction, materials and methods, results and discussion should be addressed (see comments). Thus, major changes are recommended.

Comments

1) The introduction should be focused on the growth of coppices, stand structure and diversity. Many literature references address these points in literature.

2) It is not clear if the authors made the analysis considering each pole as an individual tree or if the analysis was at stump level. Please clarify.

3) monotonous or even aged?

4) What did the authors mean by “standard forest management practices”? It can be clear for the readers of the authors' countries but not for readers of other countries.

5) What did the authors mean by “standard high forests”? It can be clear for the readers of the authors' countries but not for readers of other countries.

7) erudition or skills?

8) Please standardise the x-axis legend in the figures.

9) “crown projection area (at least in four mutually perpendicular directions) were measured”. It is not clear how many crown radii were measured

10) diameter at breast height was measured with a calliper with one or two cross measurements?

11) “the trees were divided according to classes into the upper layer (level and above-level) and the lower layer (sub-level) trees.” The results do not include the analysis at the upper and lower level.

12) Data processing section needs structuring and further details should be included in their description. All the variables, indices and methods should be described in detail.

13) The most common acronym for HDR is hd.

14) Tree ring growth and climate section needs structuring and further details should be included in their description.

15) Figure 3 does not correspond to what is written in Tree ring growth and climate section. Please clarify.

16) The scale in the axis of Figure 3 should be the same for all graphics.

17) The authors refer to a temporal analysis between 1968 and 2022, yet it is not clear the time range is in Figure 4.

18) What do the authors mean by period in Figure 5? What does the x scale represent?

19) Discussion needs structuring and should be focused on the discussion of the results. There seem to be some references missing in the text.

20) It is not clear why the authors address seed production in discussion if reproduction is mainly vegetative. Or are the analysed stands coppices with standards? If so the manuscript has to be rethought.

Comments on the Quality of English Language

Moderate editing of English language required

Author Response

The paper entitled “Turkey Oak (Quercus cerris L.) Resilience to Climate Change: Insights from Coppice Forests in Southern and Central Europe” reflects the development of applied research on the topic of coppice growth and stand structure diversity. The manuscript needs to be improved. Some issues in the introduction, materials and methods, results and discussion should be addressed (see comments). Thus, major changes are recommended.

Answer: We would like to express our sincere gratitude to the reviewer for their professional evaluation, and we greatly appreciate the time they devoted to assessing our publication. We endeavored to respond to the reviewer's comments as accurately and diligently as possible, making adjustments to the manuscript to accommodate the reviewer's suggestions to the best of our ability.

Comments

1) The introduction should be focused on the growth of coppices, stand structure and diversity. Many literature references address these points in literature.

Answer: The introduction was improved and edited to better explain coppices, biodiversity and structure subjects for this research. We also imputed more references to this subject.

2) It is not clear if the authors made the analysis considering each pole as an individual tree or if the analysis was at stump level. Please clarify.

Answer: We improved the description for methodology in chapter 2.2. We responded that “Each stem was considered as an individual tree, both for single stem and polycormons (stem with more shoots).”

3) monotonous or even aged?

Answer: The complex diversity of all plots varied between monotonous and even structure. It was clarified in Table 3 a corrected in the abstract where the was written uneven structure.

4) What did the authors mean by “standard forest management practices”? It can be clear for the readers of the authors' countries but not for readers of other countries.

Answer: It was clarified in the methodology as follows “Forest management on PRPs is based on individual thinning with respect to health sta-tus and target diameter at breast height (DBH).”

5) What did the authors mean by “standard high forests”? It can be clear for the readers of the authors' countries but not for readers of other countries.

Answer: Clarified in the Introduction and Methodology. Deleted unclearly meaning word “standard”.

7) erudition or skills?

Answer: Changed on skills.

8) Please standardise the x-axis legend in the figures.

Answer: Done.

9) “crown projection area (at least in four mutually perpendicular directions) were measured”. It is not clear how many crown radii were measured

Answer: The whole methodology for crown measurement was clarified in Data Collection.

10) diameter at breast height was measured with a calliper with one or two cross measurements?

Answer: We done two measurements that were averaged. It was inserted to the methodology.

11) “the trees were divided according to classes into the upper layer (level and above-level) and the lower layer (sub-level) trees.” The results do not include the analysis at the upper and lower level.

Answer: The methodology for structure measurement and analysis was improved and edited for an easier description of the used methodology.

12) Data processing section needs structuring and further details should be included in their description. All the variables, indices and methods should be described in detail.

Answer: Data processing was structured. We also described data processing in further details.

13) The most common acronym for HDR is hd.

Answer: Changed on hd.

14) Tree ring growth and climate section needs structuring and further details should be included in their description.

Answer: The description of the results in the mentioned chapters has been enhanced. These chapters have been more thoroughly explained and supplemented with further structuring and further details.

15) Figure 3 does not correspond to what is written in Tree ring growth and climate section. Please clarify.

Answer: The description has been edited, improved, and simplified to make it more understandable for the mentioned results in the manuscript.

16) The scale in the axis of Figure 3 should be the same for all graphics.

Answer: The axis was equalled.

17) The authors refer to a temporal analysis between 1968 and 2022, yet it is not clear the time range is in Figure 4.

Answer: The time range is “from April to December of the previous year”, and “from January to September of the current year”. However, we added to the figure caption a better description as follows „the relative year derived from the period 1968–2022“.

18) What do the authors mean by period in Figure 5? What does the x scale represent?

Answer: The description of this analysis has been given more space in the methodology of our study. We have elaborated more on the significance and individual components of this analysis. Additionally, was provided a more detailed description of what each axis represents.

19) Discussion needs structuring and should be focused on the discussion of the results. There seem to be some references missing in the text.

Answer: We improved the discussion according to these comments. The discussion was also more structured.

20) It is not clear why the authors address seed production in discussion if reproduction is mainly vegetative. Or are the analysed stands coppices with standards? If so the manuscript has to be rethought.

Answer: We explained to the discussion why was seed production mentioned. It was meant to be connected to the tree rings “Tree seed production is associated with tree-ring growth and the influence of weather in both the previous and current growing seasons [107–109]. The frequency of tree-ring growth can serve as an indicator of the theoretical fertility of trees, while we observed 2 to 5-year cycles.”

Comments on the Quality of English Language

Moderate editing of English language required

Answer: English native speaker checked the language.

Round 2

Reviewer 1 Report

Comments and Suggestions for Authors

Dear authors,

The manuscript was improved with the suggested corrections and is now suitable.

Author Response

Answer: We want to thank you for your kind and positive review.

Reviewer 2 Report

Comments and Suggestions for Authors

Journal: Forests (ISSN 1999-4907)

Manuscript ID: forests-2692878

Title: Reaction of Quercus cerris L. coppices to climate change and growth cycles from Central to Southern Europe

Overall  Comments and Suggestions for Authors

To authors,

I reviewed the resubmitted manuscript well. I found that the authors have carefully dealt with the reviewers’ comments and updated the manuscript more. I still don’t agree all parts, but some points were solved and persuaded by the authors. Moreover, the revised, updated discussions explained well about the limitations and cautions of the current study. In this way, I evaluate that the revised version can be published. One short comment to author is that it would be better to add and recommend some ideas and next steps for the following studies. I appreciate the authors’ effort on collecting the data, analyzing this study, writing the manuscript, and handling the reviewers’ comments.

Kind regards,

Reviewer

Author Response

Answer: Thank you very much for your important and useful comments on improving our article. We certainly agree with you. Newly, the last chapter in the Discussion section was named "Study limitations and ideas" and other possible trends where research should be directed in the context of adaptation to climate change, such as genetics/provenance, different silviculture methods, or the effect of admixture of other tree species, were added there. Once again, thank you very much for your professional review of our manuscript.

Reviewer 3 Report

Comments and Suggestions for Authors

The authors answered most questions in the revised manuscript. Yet, there are still they should clarify one of the terms used: monotonous and even structure are not forest terms. Please check a silviculture textbook for the correct terms. Minor changes are recommended.

Comments on the Quality of English Language

Minor editing of English language required

Author Response

The authors answered most questions in the revised manuscript. Yet, there are still they should clarify one of the terms used: monotonous and even structure are not forest terms. Please check a silviculture textbook for the correct terms. Minor changes are recommended.

Answer: Thank you very much for your professional comment. Newly, explanatory notes were added to Table 1 (see in text below). These terms are commonly used in scientific articles, e.g. Pelyukh et al. 2018, Vacek et al. 2021 and many others. But we agree with you – the term "even" was rewritten to "uniform".

We also edited notes for table that explains structure of the forest.

“Notes:

Monotonous structure = stands composed solely of a single tree species; vertically undifferentiated tree canopy; low variation in tree crown diameters; systematic spatial arrangement of trees.

Uniform structure = stands composed of 1 to 2 tree species; vertical structure of the tree canopy formed by a single layer, occasional identification of a second layer; random horizontal structure of trees.

Non-uniform structure = stands composed of up to 4 tree species with varied mixed proportions; vertical structure consisting of 2 to 3 tree layers; average crown size reaching 50%; random to weak clustering tree spatial pattern.

Diverse structure = stands composed of an average of 5 canopy-forming tree species, with 2 to 3 having similar mixture proportions; irregularly moderately multilayered vertical structure, rarely differentiated; spatial arrangement of trees classified as heterogeneous with a tendency to cluster.

Very diverse structure = forests characterized primarily by high biological diversity; vertically structured profiles forming multiple tree layers, containing up to 7 canopy-forming tree species, of which at least 3 to 4 have relatively equal representation; highly varied crown widths; spatial arrangement of trees perceived as clustered [77].”

References for this answer:

Pelyukh, O., Fabrika, M., Kucbel, S., Valent, P., & Zahvoyska, L. (2018). Modelling of secondary even-aged Norway spruce stands conversion using the tree growth simulator SIBYLA: SE “Rakhiv forestry” case study. Bulletin of the Transilvania University of Brasov. Series II: Forestry• Wood Industry• Agricultural Food Engineering, 29-46.

Vacek, Z., Prokůpková, A., Vacek, S., Bulušek, D., Šimůnek, V., Hájek, V., & Králíček, I. (2021). Mixed vs. monospecific mountain forests in response to climate change: structural and growth perspectives of Norway spruce and European beech. Forest Ecology and Management, 488, 119019.

Minor editing of English language required

Answer: English language was carefully checked more in detail by expert in the field and native speaker. Thank you very much for your comments.
